# Magnitude and factors associated with appropriate complementary feeding practice among mothers of children 6–23 months age in Shashemene town, Oromia- Ethiopia: Community based cross sectional study

Junayde Abdurahmen Ahmed[1]*, Kebede Kumsa Sadeta[1], Kelil Hussein Lenbo[2]

1 School of Public Health Shashamene Campus of Madda Walabu University, Shashemene, Ethiopia,
2 School of Nursing Shashamene Campus of Madda Walabu University, Shashemene, Ethiopia

* juneabdu@gmail.com

## Abstract

### Background

Inadequate complementary feeding is a major cause of childhood malnutrition. Malnutrition caused by insufficient complementary feeding accounts for more than one-third of all under-five mortality whereas appropriate feeding practices are critical for improving nutritional status and ensuring child survival. Thus, the objective of this study was to assess the prevalence of appropriate complementary feeding practices among mothers having 6–23 months children, from Feb.-march 2020 and associated factors in Shashemene Town, Oromia, Ethiopia.

### Methods

From February to March of 2020, a community-based cross-sectional survey was conducted. 536 mothers with children aged 6 to 23 months were chosen for the study using a two-stage sampling procedure. Data was collected by Face-to-face interviews during home-to-home visits with mothers who had children aged 6–23 months, using a structured questionnaire on the main complementary feeding indicators. The Statistical Package for the Social Sciences (SPSS) software was used to analyze the data. Logistic regression was used to identify factors associated with appropriate complementary feeding practice, with statistical significance set at probability value < 0.05.

### Results

The proportion of children aged 6–23 months who met the criteria for complementary food introduction, minimum meal frequency, minimum dietary diversity, minimum acceptable diet, and appropriate complementary feeding practices was 67.9 percent, 61.7 percent, 42.5 percent, 41.7 percent, and 30 percent, respectively. Child age 12–17 and 18–23 months were the independent factors associated with appropriate complementary feed practice

**Data Availability Statement:** All relevant data are within the paper and its Supporting Information files.

**Funding:** Madda walabu university funded budget for data collection and supervision purposes as part of employer routine activities with this reference number RMWU 14/84/355. But the funders had no role in study design, data collection and analysis, decision to publish, or preparation of the manuscript.

**Competing interests:** he authors have declared that no competing interests exist.

[adjusted odd ratio (AOR): 2.32, 95 percent confidence interval (CI): (1.40–3.82)]. ** 1.91 (1.10–3.32) **. Socioeconomic status: mothers in the wealth index of the household, second, third, and fourth, [AOR: 4.27,95 percent, CI (1.8–10.22) ** 4.02(2.23–9.94) ** 7.02 (3.27–15.1) **], number of antenatal care visits greater than or equal to four [AOR: 2.57,95 percent, CI: (1.3–5.05)] **, information sources [AOR: 3.5,95 percent, CI: (1.45–8.26) **].

## Conclusion

This study found that children aged 6–23 months had a low level of appropriate complementary feeding practice. Mothers with children aged 6–11 months, the number of antenatal care (ANC) visits, socioeconomic status, sources of information, mothers' knowledge, and positive attitude were all associated with appropriate feeding practices. As a result, nutritional education/counseling intervention on child feeding practices was suggested.

## Introduction

Malnutrition is still one of the leading causes of morbidity and mortality among children worldwide. It has been directly or indirectly responsible for 60 percent of the 10.9 million deaths among children under the age of five each year, with two-thirds of these deaths frequently associated with inappropriate feeding practices [1, 2]. Children are particularly vulnerable to malnutrition in developing countries due to inadequate dietary intakes, a lack of appropriate care, and inequitable food distribution within households [3, 4].

Complementary feeding is the gradual introduction of solid and semisolid foods into an infant's diet when breast milk alone is insufficient to meet the nutritional needs of the infant. The age range for complementary feeding is considered to be 6 to 23 months, despite the fact that breastfeeding may continue for up to 2 years [3, 5, 6].

Inadequate complementary feeding practices have been identified as a global risk factor for diarrheal diseases, malnutrition outcomes, and under-5 mortality [7]. Only one in every six children worldwide receives a minimally acceptable diet [8]. According to a study conducted in South Asia, there are significant differences in complementary feeding practices. For example, India had the lowest proportion 15% of children 15% who met the minimum dietary diversity (MDD), followed by Nepal (34%), Bangladesh (42%), and Sri Lanka (71%) [9, 10]. One out of every ten infants and young children received minimum acceptable diet in Eastern and Southern Africa [8]. According to the EDHS (2016) [11], only 7% of children aged 6–23 months met the criteria for a minimum acceptable diet, the lowest ever; Ethiopia has taken various steps in the implementation of the Integrated Child Development Scheme to improve children's nutritional status [12]. However, evidence suggests that appropriate complementary feeding practices are still low. According to a study conducted in Nagele Arsi Oromia Region, Ethiopia the prevalence of appropriate complementary feeding practices was 9.5 percent, the minimum acceptable diet was 12.3 percent [13] and Damota sore of south Ethiopia was 11.4 percent [14].

The World Health Organization (WHO) created eight core and seven optional indicators to monitor and guide infant and young child feeding practices [5, 6]. Ethiopia, a Sub-Saharan African country with a high level of malnutrition, launched a national strategy for infant and young child feeding in 2004 to improve children's nutritional status [12].

Concerning factors associated with appropriate complementary feeding practices of children aged 6–23 months previous studies conducted elsewhere show higher maternal and paternal education, better household wealth, exposure to media, adequate antenatal and postnatal contacts, child's sex and age, institutional delivery, low parity, maternal occupation, urban residence, knowledge & frequency of complementary feeding and receiving feeding advice about immunization was determinant factors for appropriate complementary feeding [9, 10, 13–17].

Appropriate feeding practices are critical for improving nutritional status and ensuring child survival thus, the WHO recommends a combination of indicators to measure the level of appropriate complementary feeding but most studies on complementary feeding practices to date have used a single indicator or three indicators with a narrow age range, failing to adequately quantify the level and determinants of appropriate complementary feeding practices [5, 6].

Malnutrition is linked to inappropriate feeding practices that can be avoided. Children aged 6–23 months are particularly vulnerable, and it is also the time when malnutrition begins in many infants, contributing to the high prevalence of malnutrition in children under the age of two plus, Ethiopia particularly, in Shashamene there were diversified ethnic groups that have different cultures, beliefs or feeding habit or customs affects child feeding. Many studies had been done in rural part of Ethiopia.

Beside this, currently EBF practices were also reduced in Ethiopia [11]. Moreover, Infant feeding practices change over time, and place. Apart from that, there is a lack of information on appropriate complementary feeding in developing countries, which applies to the study setting in order to explain the level of appropriate complementary feeding and associated factors. However, it has never been assessed in Shashemene town.

Hence, knowing the level of practices helps to undertake appropriate measures to improve child feeding practice that in turn improve nutritional status of children through this essential time of child growth and development. As a result, the study aimed The objective of this study was to assess the prevalence of appropriate complementary feeding practices among mothers having 6–23 months children, from Feb.-march 2020 and associated factors in Shashemene town. The findings will aid in the promotion of appropriate complementary feeding and will serve as baseline data for any concerned bodies.

## Material and methods

### Study settings and period

Shashemene is the most densely populated town in the Oromia region of Ethiopia, with a diverse ethnic population. It is located 250 kilometers from Addis Ababa, Ethiopia's capital city. Shashemene town is located in the subtropical climatic zone: In 2019, the population of Shashemene town is estimated to be 272193, with 50.4 percent males and 49.6 percent females. According to the 2020 Shashemene Town report [18], children aged 6–23 months made up 4.8 percent of the population, or 13065 people. The research was carried out in Shashemene Town from February to March 2020.

### Study design and study population

From February to March 2020, a community-based cross-sectional study was conducted in Shashemene, Oromia, Ethiopia. All mothers with children aged 6–23 months who lived in Shashemene town by 2020 were considered the source population, whereas mothers with children aged 6–23 months who lived in selected households during the study period and lived in the study area for more than 6 months were considered the study population.

Mothers who resided in the study area for <6 months were excluded from the study subjects.

## Inclusion and exclusion criteria

The source population was all mothers-child pairs aged 6–23 months living in Shashamene town. Mother-child pairs aged 6–23 months living in selected households during the study period as well as those residing in the study area for 6 months presented during the study period were included as study subjects.

## Independent and dependent variables

**Dependent variable was expressed as:** Mothers' complementary feeding practices which the response can be dichotomized and coded as [1 = Appropriate complementary feeding practice, 0 = inappropriate complementary feeding practice].

**Appropriate complementary feeding practice:** defined appropriate when they meet all the four Complementary feeding indicators timely introduction, minimum meal frequency and minimum dietary diversity and minimum acceptable diet and coded as 1 while it is considered inappropriate complementary feeding practice when it fails to fulfill even a single indicator [5, 6].

**Timely introduction of complementary feeding:** The proportion of children 6–23 months that were introduced to solid and semisolid foods at 6 months of age [5, 6].

**Minimum dietary diversity:** is the proportion of children 6–23 months of age who receive foods from 4 or more food groups with the food groups consisting; (I) grains, roots and tubers; (II) legumes and nuts; (III) dairy products; (IV) flesh foods; (V) eggs; (VI) vitamin A rich fruits and vegetables; and (vii) other fruits and vegetables during the previous day of study [5, 6].

**Minimum meal frequency:** is the proportion of breastfed and non-breastfed children 6–23 months of age, who receive solid, semi-solid, or soft foods (but also including milk feeds for non-breastfed children) the minimum number of times or more during the previous day.

**Minimum** is defined as 2 times for breastfed infants 6–8 months, 3 times for breastfed children 9–23 months, 4 times for non-breastfed children 6–23 months [5, 6].

**Minimum acceptable diet:** is the proportion of children 6–23 months of age who receive both minimum meal frequency and minimum dietary diversity during the previous day of study [5, 6].

**Knowledge and attitude on appropriate complementary feeding practice (ACFP)** was measured among respondents using

- **10 items** [(1). ever breast feed your child? (2). Time of initiation of first breast milk? (3) How long you gave Breast Milk for the baby? (4) having information on CF; (5). have knowledge about the advantage of iron rich food? (6). know the benefit of Iodized salt? (7). frequency of feeding solid, semisolid or soft food] (8) sources information about advantage of breast feeding? (9). sources information about commercially prepared complementary food? and (10). how to prepare complementary foods? and

- **4items** [1. It is important to help my child when she eat, 2. It is important to feed my child to eat slowly and patiently,3. It is important to encourage my child to eat and 4. I talk to my child during feeding by looking straight in the eyes] respectively. For each item, mothers responded

- „„„Yes were given a value of „„1, while those cited „„No were given „„0 and summed up. Those mothers scored mean or more were considered as having good knowledge, whereas those scored below mean value were characterized as poor knowledge similarly procedure

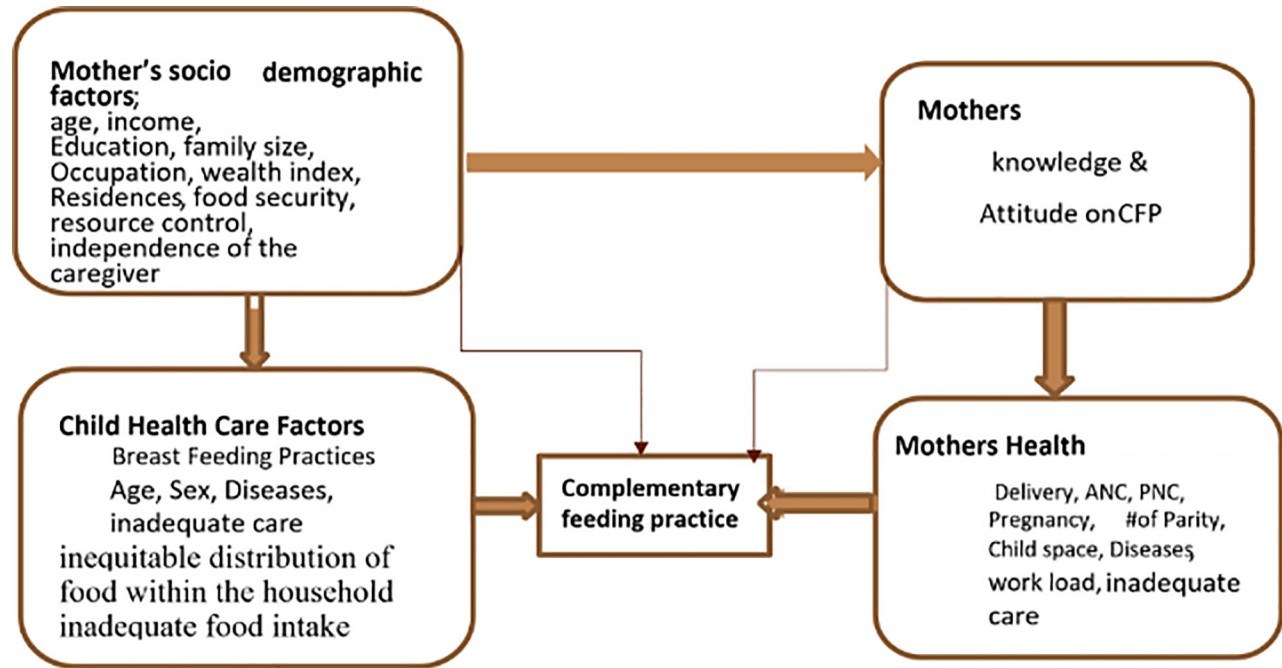

**Fig 1. Conceptual frame work of complimentary feeding practice from different literature review, 2020.** CFP: Complementary feeding practices, #: number, ANC: Antenatal care, PNC: Postnatal care.

were followed for attitude those who scored mean or more value were considered as having good attitude, whereas those scored below mean value were categorized as poor attitude towards ACFP [Fig 1]. Conceptual frame work of complementary feeding practice from different literature review.

- **Maternal, child, and household characteristics were the independent variables**. The variables were described in brief as follows: Socioeconomic characteristics: -age, gender, family size, monthly income, partner education level, household wealth index, food insecurity, occupation, residence, knowledge, and attitude of mothers Obstetric history (Pregnancy History, ANC, PNC, delivery mode and place of delivery, birth space, and number of Parity) were considered independent variables, while ACFP was considered dependent.

Household food security was measured with the Household Hunger Scale (HHS) which has 9 items along with 9 frequencies (9I 9F) Household Food Insecurity Access Scale (HFIAS) [18]. The household food insecurity status was measured by direct survey of household consumption 4weeks preceding the survey. In this study, household food insecurity is dichotomous variable taking value 1 if the household is food insecure and 0 otherwise. The response categories are never (0 times), rarely (1–2 times), or sometimes (3–10 times), and often (more than 10 times). Therefore, the HHS was used in this study to define two groups; households reporting (a) little to no hunger in the past month because of insufficient food or because of lack of resources to get food and thereby classified as food secure households, and (b) moderate to severe hunger in the past month because of insufficient food or because of lack of resources to get food, and thereby classified as food insecure households.

## Sample size determination

The sample size was calculated using a single population proportion formula considering the proportion of appropriate complementary feeding practice (11.4%) from a previous study [14].

$$n_o = \frac{z\alpha/_2^2 p(1-p)}{d^2} \ggg \text{n} = \frac{(1.96)^2 * 0.114(1-0.114)}{(0.04)^2} = 243, 243*2+48 = \underline{\textbf{536}}$$

The following assumptions were used:

- Margin of error = 4%;

- Zα = 1.96 and

- Design effect = 2.

243 samples were obtained with consideration of 10% contingency to non-responders a total of 536 mothers were sampled

## Sampling procedure

A two stage sampling technique was used to select the study subjects. Four (4) sub cities was randomly selected using simple random sampling method from 8 sub cities. The total population in the four (4) selected sub cities: Bulchana, Arada, Alelu and Awasho was 139510 (respectively 36877, 34529, 31734 and 36370 of which 6696 was children of 6–23 months of age.

The calculated sample (536) was allocated equally among the selected 4 sub cities i.e. 134 mothers having children 6–23 months in each sub city. To select the individual sample units or subjects at household level, all target groups at each sub city was obtained from the health post then K$^{th}$ was calculated.

The random start was determined by lottery, and every Kth mother with eligible children was chosen from four sub-cities using systematic random sampling, so a child was chosen in each sub-city and his or her mother was interviewed accordingly. From each household, one eligible child with a mother at the time of the survey was chosen; if more than two eligible children were found, the younger was chosen, and the process was repeated until the next Kth in the same direction. If the mother was not present on the date of data collection, she was replaced by the next mother from the same sub city after one revisit [Fig 2]. Diagrammatic presentation of sampling scheme of the sampled mothers-child pairs of 623months age Shashamene, Ethiopia, 2020.

## Data collection instruments

The questionnaire was created by reviewing various literatures and then adapting it to the local context [5, 6, 13, 14]. Face-to-face interviews were conducted during home-to-home visits with mothers who had children aged 6–23 months, and data were collected using a structured questionnaire. The questionnaire included questions about the mothers and children's backgrounds, maternal health practices, and child feeding practices.

## Data collection methods

Face-to-face interviews were conducted during home-to-home visits with mothers who had children aged 6–23 months, and structured questionnaires were used to collect data. The questionnaire included questions about the mothers and children's backgrounds, maternal health practices, and child feeding practices. Six diploma holders nurses were hired as data collectors,

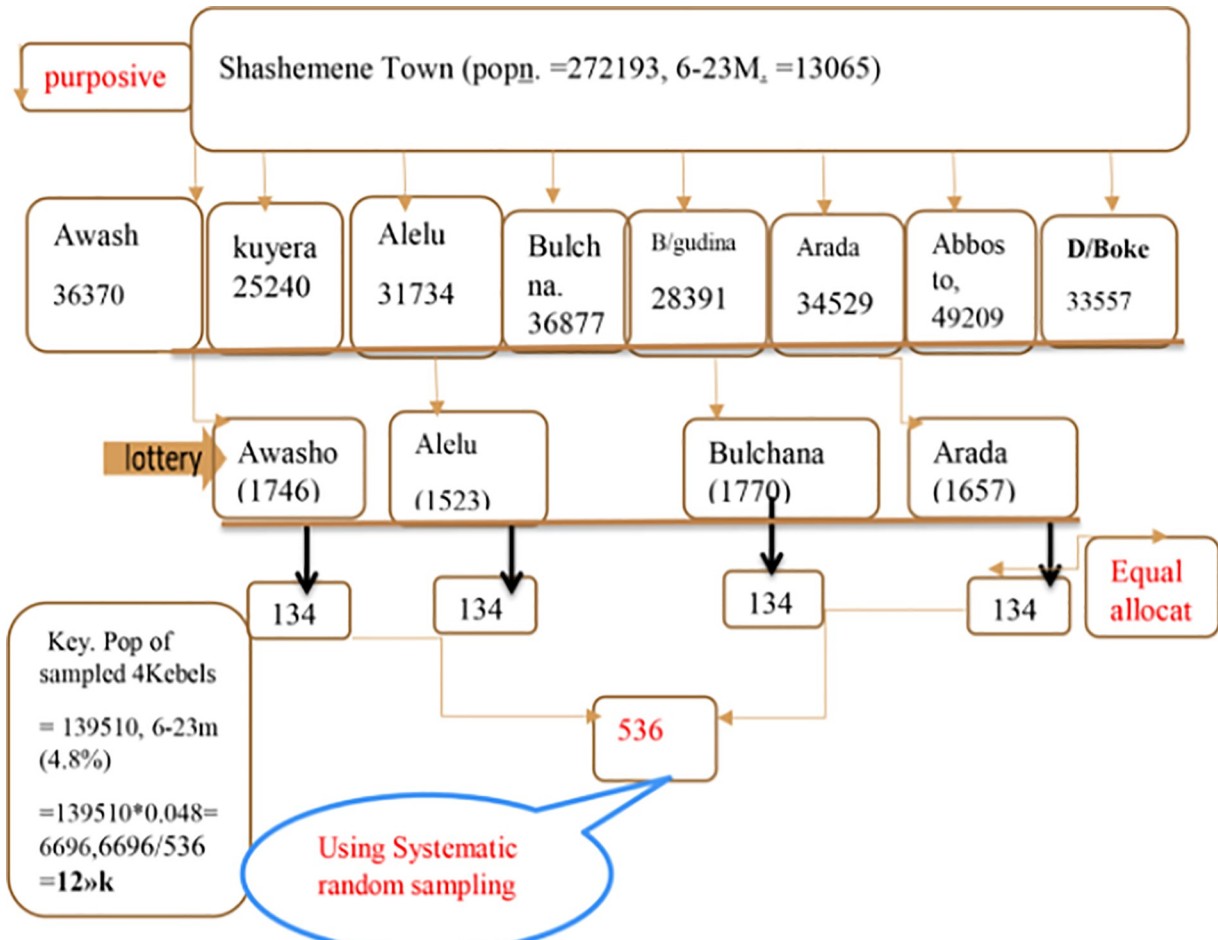

**Fig 2. Diagrammatic presentation of sampling scheme of the sampled mothers-child pairs of 6–23 months age Shashemene, Ethiopia, 2020.** D/Boke: Didaboke, pop(n): Population, B/gudina: Burka Gudina, m:month.

and two BSc holders were hired as supervisors. For data quality control, the questionnaire was first written in English, then translated into the local language, Afaan Oromo, and then back translated into English by two people who are fluent in both languages. Data collectors and supervisors were trained for two days, and the questionnaire was pre-tested 26 mothers in the study area who were not included in the actual study to assess the content and approach of the questionnaire, and necessary corrections were made. All questionnaires were checked for completeness on a daily basis, and data was thoroughly checked and cleaned before analysis. The questionnaires were created by reviewing various literatures and validating them in the context of our country.

## Data processing and analysis

The statistical package for social science (SPSS) version 25 was used to code, enter, and analyze data. Data was described using descriptive statistics such as frequencies, proportions, means, and standard deviation. Bivariate analysis was performed to appreciate the relationship between each independent variable and the dependent variable. Finally, independent variables associated with P-value 0.25 during bivariate analysis were entered into multivariable logistic regression analysis, which was used to determine the strength of association between

independent and dependent variables. Odds ratios with 95 percent confidence intervals were reported, and statistical significance was declared at a p-value of 0.05. To control for confounders, a multivariate logistic regression model was used. To handle the effect of residual confounding effect we took representative sample size. Beside this we used standardized questionnaire to include all variable that are important to measure ACFP as well we tried to control confounding with multivariate analysis.

Using principal component analysis, the wealth index was calculated as a measure of household wealth (PCA). Fifteen variables were considered, including ownership of selected household assets, the size of the quantity of durable equipment, materials used in housing construction, and ownership of improved water and sanitation facilities. Finally, the resulting principal component was divided into five equal quintiles (lowest, second, middle, fourth, and highest).

## Ethical considerations

The Madda Walabu University (MWU) research and ethics committee reviewed and approved the research proposal. Permission was obtained from the appropriate authorities. The consent was in accordance with the ethical principle of "autonomy" statements, which give participants the right to stop/ withdraw from the study at any time. 'before we collected data from each individual, information about informed verbal consent was provided then as the study was not invasive in nature the informed consent was obtained from each study participants or the parent or guardian. In short, the information sheets were available on each questionnaire then, we provided information about the purpose, procedures, benefits and disadvantage of the study for the participants or guardians. Informed due to the nature of the study was not invasive. Lastly, only participants who were willing to participate in the study were included. Moreover, those who wish to stop their participation at any stage were also permissible to do so without any restriction'. Finally, interviewers will inform respondents about the importance of appropriate CF practice and nutritional advice. In addition, intervention strategies will be developed in order to take appropriate action.

## Result of the study

### Socio demographic characteristics of study subjects

From the 536 sampled mothers, 520 mother-child pairs participated in the study, yielding a response rate of 97.01 percent. The mothers' average age was 26.83 SD (+4.41) years. More than half of the study participants were between the ages of 25 and 34. In terms of educational attainment, approximately 22 percent of mothers had only a basic education, while 14.8 percent of households did not work. Of the total study subjects, 481 (92.5 percent) were married, 14 (2.7 percent) lived in different places and were not married, 10 (1.9 percent) were divorced, and 1 was widowed (.2 percent). 325 (62.5%) Muslims by religion and More than half of the 333 respondents (64 percent) were Oromo [Table 1].

The majority of study participants (67.5 percent) were multiparous, and (95 percent) received antenatal care. Four hundred fifty (86.5 percent) of respondents had a birth interval of more than two years from their previous birth (Table 2).

Almost all mothers (519, or 99.8%) BF their children after delivery, indicating that very young children are mostly fed breastmilk, as recommended. 496/520 (or 95.5 percent) began BF earlier. Specifically, 272 (52.4 percent) give breast milk immediately after birth, 224 (43.1 percent) within one hour, the remaining 23 (4.4 percent) within 24 hours, and the majority of them (482 92.7 percent) feed breast milk based on child demand.

**Table 1. Socio demographic characteristic of mothers with children 6–23 month, Shashemene town, Ethiopia, 2020 (n = 520).**

| Age of mothers/caretakers | Frequency | percent |
|---|---|---|
| < = 24 years | 151 | 29.4 |
| 25–29 years | 231 | 44.4 |
| 30–34 years | 100 | 19.2 |
| 35+ | 38 | 7.3 |
| **Religion of mothers** | | |
| Muslim | 325 | 62.5 |
| Orthodox | 95 | 18.3 |
| Protestant | 81 | 15.5 |
| Catholic | 13 | 2.5 |
| Others | 6 | 1.2 |
| **Educational status of Mothers** | | |
| Basic Education | 117 | 22.5 |
| Primary school | 236 | 45.4 |
| Secondary school | 113 | 21.7 |
| College /university | 54 | 10.4 |
| **Educational status of Fathers** | | |
| Basic Education | 71 | 13.7 |
| Primary school | 189 | 36.3 |
| Secondary school | 192 | 36.9 |
| Higher Education | 68 | 13.1 |
| **Mothers/caretakers occupation** | | |
| Housewife | 244 | 46.9 |
| Daily Labor | 150 | 28.8 |
| Gov't employee | 64 | 12.3 |
| Students | 18 | 3.5 |
| Merchant | 44 | 8.5 |
| **Husband's occupation** | | |
| Gov't employee | 149 | 28.7 |
| Farmer | 77 | 14.8 |
| Merchant | 217 | 41.7 |
| Daily laborer | 28 | 5.4 |
| Others | 49 | 9.4 |
| **Family income per month** in Ethiopian Birr(ETB) | | |
| < = 999 ETB | 13 | 2.5 |
| 1000–1999 ETB | 84 | 16.2 |
| 2000–2999 ETB | 129 | 24.8 |
| 3000–3999 ETB | 89 | 17.1 |
| > = 4000 ETB | 205 | 39.4 |
| **Ethnicity** | | |
| Oromo | 333 | 64 |
| Amhara | 89 | 17.2 |
| Wolaita | 47 | 9 |
| Others | 51 | 9.8 |
| **Family size** | | |
| 1–3 | 101 | 19.4 |
| 4–6 | 327 | 62.9 |

(*Continued*)

**Table 1.** (Continued)

| Age of mothers/caretakers | Frequency | percent |
|---|---|---|
| > = 7 | 92 | 17.7 |
| **Who decide on the properties of the Household** | | |
| Husband | 166 | 31.9 |
| Wife | 31 | 6 |
| Jointly | 323 | 62.1 |
| **Sources of information about commercially available CF** | | |
| HCWs | 95 | 18.3 |
| Family | 163 | 31.3 |
| Media | 215 | 41.3 |
| Relative | 16 | 6.9 |
| Others | 11 | 2.1 |
| **Sources of information about Advantage of Breast Milk** | | |
| TV | 77 | 14.8 |
| Radio | 97 | 18.7 |
| Health professional | 237 | 45.6 |
| Reading | 54 | 10.4 |
| Family/relative | 55 | 10.6 |
| **Knowledge about advantage of iodized salt** | | |
| Yes | 365 | 70.2 |
| No | 155 | 28.8 |
| **Knowledge about advantage of Iron rich food** | | |
| Yes | 267 | 51.3 |
| No | 253 | 48.7 |
| **Source of drinking water** | | |
| Piped into dwelling/Public tap | 505 | 97.1 |
| Bottled water | 15 | 2.9 |
| **Household wealth index** | | |
| Lowest | 13 | 2.5 |
| Second | 61 | 11.7 |
| Middle | 203 | 39 |
| Fourth | 151 | 29 |
| Highest | 92 | 17.7 |
| **House Hold food security** | | |
| Secured food | 407 | 78.3 |
| In secured | 113 | 21.7 |

ETB = Ethiopian Birr, TV: Television, CF: complementary feeding, HCWs: health care workers.

**Note**: Table 1 Socio demographic characteristic of mothers with children 6–23 month.

Thirty-one (6%) mothers begin CF earlier. Mothers' negative attitude toward the quantity of breast milk, failure to stay with the child, and a lack of knowledge about breast milk were the most common reasons for early initiation of CF. in Ethiopia, the median and mean duration of any BF are both 25 months. Fig 3 depicts the mothers' BF status; the median duration of BF was 14 months. I.e. 243/520 (46.7%) lower than the EDHS, 2011 [Fig 3]. Indicator of breast feeding status among 6–23 month age children in Shashemene oromia, Ethiopia, 2020.

**Table 2. Maternal obstetric related characteristics of mothers of children 6–23 month Shashamene town, Oromia-Ethiopia, 2020.**

| Attributes | Nᵒ | Percent |
|---|---|---|
| **Parity** | | |
| Prim parous (1) | 92 | 17.7 |
| Multiparous (2–4) | 351 | 67.5 |
| Grand multipara (5+) | 77 | 14.8 |
| **Delivery attendants** | | |
| HCW | 460 | 88.5 |
| TBAs' | 35 | 6.7 |
| Other's | 25 | 4.8 |
| **Mode of delivery** | | |
| Normal | 468 | 90 |
| Cesarean Section (C/S) | 49 | 9.4 |
| Others | 3 | 0.6 |
| **Place of delivery** | | |
| Home | 94 | 18.1 |
| Health Facilities | 426 | 81.9 |
| **Nᵒ ANC (Antenatal care) attendants** | | |
| No ANC session | 26 | 5 |
| < = 3 | 279 | 53.7 |
| > = 4 | 215 | 41.3 |
| **Attend ANC services** | | |
| Yes | 494 | 95 |
| No | 26 | 5 |
| **Attend Post-natal care (PNC)** | | |
| Yes | 288 | 55.4 |
| No | 232 | 44.6 |
| **Time of PNC attendance** | | |
| Not attend | 231 | 44.4 |
| 0-2days | 128 | 24.6 |
| 3-6days | 127 | 24.4 |
| 7days & above | 34 | 6.5 |
| **Birth Spacing** | | |
| <2 | 338 | 65 |
| 2–4 | 160 | 30.8 |
| >4 | 22 | 4.2 |
| **Nᵒ of Children in the family** | | |
| 1–3 | 384 | 73.8 |
| 4–6 | 120 | 23.1 |
| >6 | 16 | 3.1 |
| **Birth order** | | |
| 1st | 101 | 19.4 |
| 2nd | 141 | 27.1 |
| 3rd | 152 | 29.2 |
| 4th | 59 | 11.3 |
| 5th &above | 67 | 12.9 |
| **Sex of the child** | | |
| Male | 262 | 50.4 |

(*Continued*)

**Table 2.** (Continued)

| Attributes | N° | Percent |
|---|---|---|
| **Parity** | | |
| Female | 258 | 49.6 |
| **Age of the child** | | |
| 6-8Months | 93 | 17.9 |
| 9-11Months | 70 | 13.5 |
| 12-17Months | 209 | 40.2 |
| 18-23Months | 148 | 28.5 |
| **Did the child have diarrhea in the last 2Week?** | | |
| Yes | 239 | 46 |
| No | 281 | 54 |
| **Did the Child had acute respiratory tract infection (ARTI)** | | |
| Yes | 292 | 56.2 |
| No | 228 | 43.8 |
| **Did the Child had Fever** | | |
| Yes | 329 | 63.3 |
| No | 191 | 36.7 |

TBAs = Traditional Birth Attendants, TBAs = Traditional Birth Attendants, HCW = health care workers, ANC (Antenatal care), C/S = Cesarean Section.

**Note**: Table 2 Maternal obstetric related characteristics of mothers of children 6–23 month old age.

## Food and fluid provided for the children

Milk of any type (417(80.2 percent) was the most commonly provided food or fluid for children, followed by potatoes (324(62.3 percent), porridge/gruel 235(45.2 percent), and bread, vegetables, and fruits 181 (34.8 percent). Fed with foods from four or more of the following groups: **a**. infant formula, milk other than breast milk, and cheese, yogurt or other milk

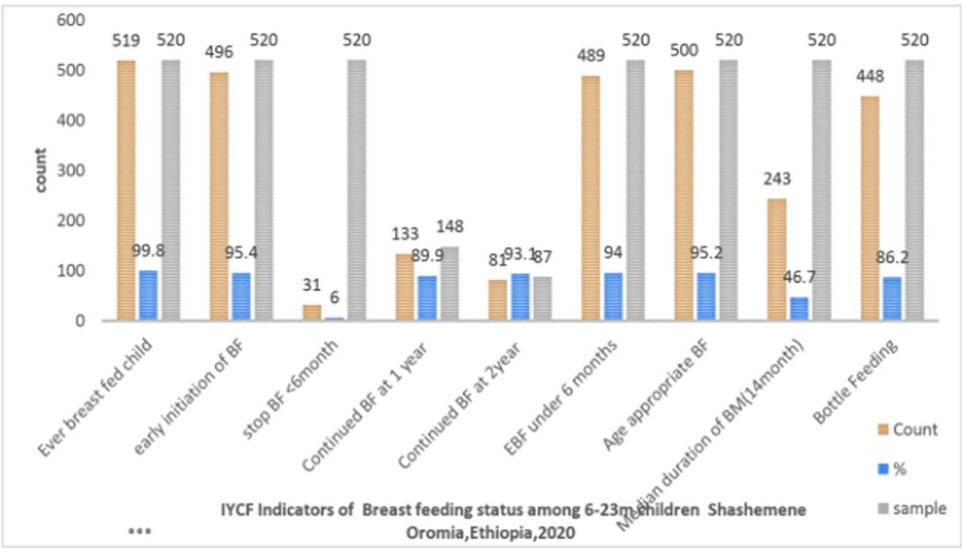

**Fig 3. Indicator of breast feeding status among 6–23 month age children in Shashemene Oromia, Ethiopia, 2020.**
BF: Breast feeding, IYCF: Infant and young child feeding, BM: breast milk, EBF: exclusive breast feeding.

**Table 3. Types of food groups given during the previous day according to age of the child among children aged 6–23 months, Shashemene, 2020 (N = 520).**

| Food groups | Age of children in months Nº(%) | | | | | |
|---|---|---|---|---|---|---|
| | 6–11 months (N = 163) | | 12–17 months (N = 209) | | 18–23 months (N = 148) | |
| | Yes | | Yes | | Yes | |
| | Nº | (%) | Nº | (%) | Nº | (%) |
| Grains, roots & tubers | 135 | **82.8** | 187 | **89.5** | 131 | **88.5** |
| Legumes and nuts | 65 | **39.5** | 94 | **45** | 84 | **56.8** |
| Dairy products | 107 | **65.6** | 139 | **66.5** | 91 | **61.5** |
| Egg | 71 | **43.6** | 110 | **52.6** | 90 | **60.8** |
| Flesh foods | 20 | **12.3** | 39 | **18.7** | 39 | **26.4** |
| Vit. A rich foods | 90 | **55.2** | 116 | **55.6** | 108 | **73** |
| Other fruits & vegetable | 77 | **47.2** | 75 | **35.9** | 73 | **49.3** |

**Vit = vitamin A.**

**Note**: Table 3 Types of food groups given during the previous day according to age of the child among children aged 6–23 months.

products; **b.** foods made from grains, roots, and tubers, including porridge and fortified baby food from grains; **c.** vitamin A-rich fruits and vegetables; **d.** other fruits and vegetables; **e**. eggs; **f.** meat, poultry, fish, and shellfish (and organ meats); and **g.** legumes and nuts. Homemade food was preferred by 398 (76.5 percent) of the children. The proportion of mothers who preferred commercially available food for their children was 122 (23.5 percent). In terms of CF practices and frequency during illness, 187 (36%) mothers reduced the quantity and the frequency of food, 147 (28.3%) withheld the quantity and frequency of food, 123 (23.7%) maintained the same quantity and frequency, and only 63 (12%) mothers used to increase the quantity and the frequency of food. Despite cultural and social food restrictions, the most commonly consumed foods are cabbage and meat, which account for 124 percent of total consumption (23. 8 percent). In terms of food preparation, 452 (86.9 percent) of the 520 mothers prepare separately for their children, while 68 (13.1 percent) prepare with adult food.

Grains, roots and tubers, dairy products, and vitamin A-rich foods were the most commonly consumed food items by children of all ages in the 24 hours preceding the survey. In the 24 hour preceding survey, children aged 18–23 months consumed more legumes and nuts, eggs, and vitamin A-rich fruit and vegetables than the other groups. In comparison to other food groups, eggs were the least consumed in the age group 6-11months; however, there was less consumption of flesh food across all age groups (Table 3).

## Complementary feeding status' indicators

CF was assessed on 520 mothers-child pairs who participated in the study. Three hundred fifty-three (67.9 percent) of the mothers were introduced for CF at 6 months' age of the children, as suggested, while 31(6 percent) were introduced before 6 months and the rest were introduced late after 6 months. Introduction of solid, semi-solid, or soft foods at the age of 6-8months was 86/93 (92.5 percent), age 12-17month 112/209 (53.6 percent), age 9-11month 53/70 (75.7 percent), age 12-15month(68.9 percent), and 18-23m (68.9 percent) which varies across different age categories.

Overall, 321 (61.7 percent) of mothers who fed their children MMF as recommended. Overall, less than half (41.7%) of children aged between 6–23 months met the requirement for MAD. The highest proportion of mothers who fed their children MAD was in the age group of

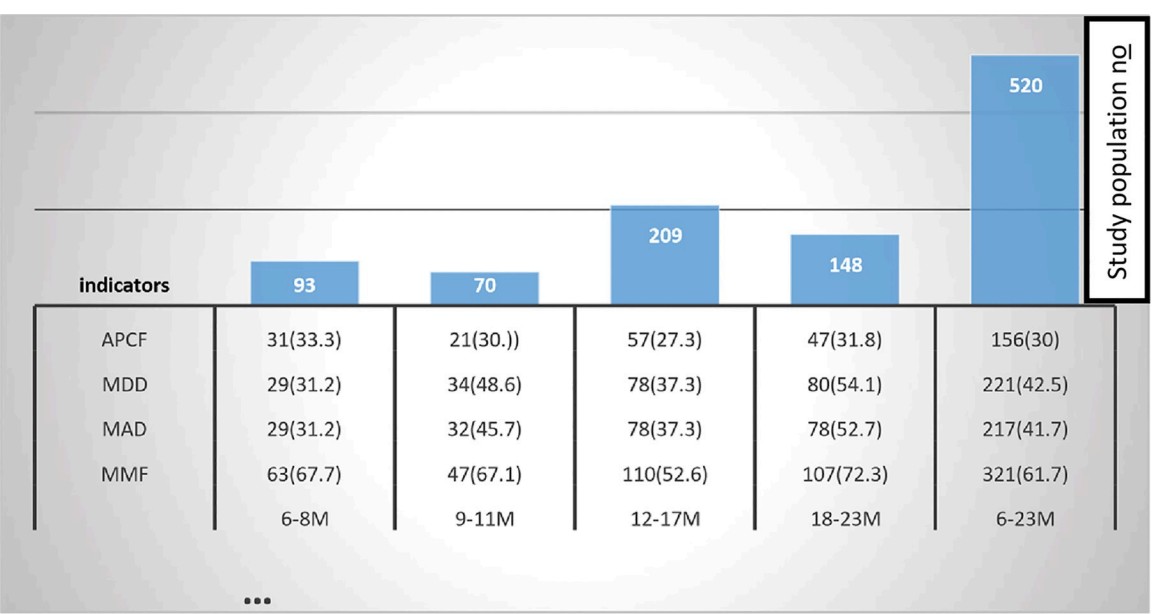

**Fig 4. Mothers' children feeding practice status with different indictors by age n (%) Shashemene Ethiopia, 2020.** APCF: appropriate complementary feeding practice, MAD: minimum acceptable diet, MDD: minimum dietary diversity, MAD: minimum acceptable diet, and MMF: Minimum meal frequency, n(): count,() = percentage, M: age in months. Fig 4 depicts the status of mothers' children's feeding practices in relation to various indicators such as appropriate complementary feeding practice (APCF), minimum dietary diversity (MDD), minimum acceptable diet (MAD), and Minimum meal frequency (MMF) in the age groups 6–8 months, 9–12 months, 12–17 months and 18–23 month with corresponding number and percentages of children in Shashemene Ethiopia, 2020.

18–23 months which was 78(52.7%). The lowest proportion of mothers who fed their children MAD was in the age group of 6–9 months which was 31%. By combining the four indicators, the ACFP level was 156 (30 percent) [Fig 4].

Slightly more than half of the children aged 9–23 months that was 112/190 (53·1%), 256/399, (64.2%), and 260/399 (65.2%). met ACFP, MDD, and MAD criterion for BF Children respectively. Eighty three 83 /93 (89.2%) mothers fed their children one or two times among breastfed child 6–8 months' age whereas 313(73.3%) for breast-fed child 9-23months age fed three or more times.

The proportion of BF mothers with child age of 6-23months who fed their children and met the MMF, MDD, and MAD criteria were 376/489(76.9%), 261/489, (53.4%), and 327/489, (66.9%) respectively. Similarly, the proportion of BF mothers with 9 to 23 months who met MMF was 307/399(76.9%). Nearly more than ¾ of children aged 6–8 months (76.7%), had met MMF for BF children. Of 31 Non BF children of age 6-23months only 21/31(67.7%), 14/31 (45.2%),10/31(32.3%) met the criteria for MMF, MDD and MAD respectively.

## Knowledge and attitude toward ACFP

Overall, 319 (61.3 percent) of respondents knew a lot about ACFP. In terms of attitude, approximately 34 of 391 participants (75.2 percent) had a positive attitude toward ACFP.

**Bivariate and multivariable analyses.** Table 4 shows factors associated with ACFP among mothers having children aged 6–23 months were identified: A Variables with P. value of less than 0.25 in bivariate analyses were maternal education, occupation, parity, birth interval, family size, household income, child age, place of delivery, maternal ANC, Number of ANC, PNC, sources of information, wealth index, food security issues, knowledge and attitude were the variables studied. In this study no association was found between maternal occupation and education, birth interval, family size, ANC, place of delivery, PNC and ACFP.

**Table 4. Factors associated with complementary feeding of children age 6–23 months, Shashamene -Oromia, Ethiopia, 2020.**

| Variables | Complementary feeding | | COR (95% CI) | AOR (95% CI) |
|---|---|---|---|---|
| | Appropriate | Inappropriate | | |
| **Maternal education** | No (%) | No (%) | | |
| No formal education | 39(25) | 78(21.4) | 1 | 1 |
| Primary school | 73(46.8) | 163(44.8) | 0.90 (.57–1.43) | .92(.57–1.48) |
| ≥2$^{ry}$ school | 63(34.2) | 104 (31) | 1.08 (.67–1.77) | 1.27 (.076–2.13) |
| Maternal occupation | 87(55.8) | 157 (43.1) | 1 | 1.00 |
| Housewife | | | | |
| Daily Labor | 32(20.5) | 118 (32.4) | 1.26(.82–1.92) | 0.82(0.37–1.81) |
| Gov't | 16(10.3) | 48(13.2) | 1.50(.82–2.63) | 1.94(.82–4.57) |
| Students | 9(5.8) | 9(2.5) | .59(.19–1.84) | 1.58(.60–4.22) |
| Merchants | 12(7.7) | 32(8.8) | 1.17(.60–2.29) | |
| Sex of the child | | | | |
| Male | 72(46) | 190(52.2) | 1.04(.73–1.50) | .77(1.05–74) |
| Female | 84(53.8) | 174(47.8) | 1 | 1 |
| Family size | 40(21.7) | 61 (18.2) | 2.37 (0.89–6.33) | 0.78 (.29–2.06) |
| 1–3 | | | | |
| 4–6 | 104 (56.5) | 223 (66.4) | 2.68 (1.17–6.13) | 0.88 (0.38–2.44) |
| ≥7 | 40(21.7) | 52(15.5) | 1.00 | 1.00 |
| Family income/monthly | 6(3.8) | 7 (1.9) | 6.32 (0.78–51.10) | 6.58 (.79–54.53) |
| ≤999 ETB | | | | |
| 1000–1999 ETB | 22(14.1) | 62 (17) | 5.59(.70–44.46) | 7.19(0.88–58.72) |
| 2000–2999 ETB | 44 (28.2) | 85(23.4) | 8.15(1.02–65.47)* | 10.10 (1.21–84.19)* |
| 3000–3999 ETB | 27(17.3) | 62(17) | 7.22 (0.92–56.60) | 10.02(1.23–81.26)* |
| 4000+ ETB | 57(36.5) | 148 (40) | 1.00 | 1.00 |
| Age of the child | 40(21.7) | 53 (15.8) | 1 | 1.00 |
| 6–8 months | | | | |
| 9–11 months | 25 (38.6) | 45 (13.4) | 1.39 (.82–2.37) | 1.40 (0.77–2.56) |
| 12-17months | 67(36.4) | 142 (42.3) | 2.75 (1.2–11.80) * | 2.32, (1.40–3.82) ** |
| 18–23 months | 52(28.3) | 96 (28.6) | 1.81(1.15–2.85) ** | 1.91(1.10–3.32)** |
| Place of delivery | 118 (75.6) | 308(84.6) | .83(.52–1.33) | 1.18(.39–2.47) |
| Health facility | | | | |
| Home | 38 (24.4) | 56 (15.4) | 1.00 | 1.00 |
| No of ANC attendance | | | | |
| No ANC | 7(4.5) | 19(5.2) | 1 | 1 |
| ≤3 | 62(39.7) | 217(59.6) | 1.22(0.61–2.40) | 1.14(0.57–2.29) |
| > = 4 | 87(55.8) | 128(35.2) | 2.72(1.40–5.29)** | 2.57(1.3–5.05)** |
| Attended ANC | 56 (98.2) | 494(89.2) | 6.80 (0.92–50.03) | 0.82(.33–2.03) |
| Yes | | | | |
| No | 1 (1.8) | 60 (10.8) | 1.00 | 1.00 |
| Attended PNC | 86 (55.1) | 202(55.5) | .96 (.67–1.38) | 1.04 (0.69–1.358) |
| Yes | | | | |
| No | 70(44.9) | 162(44.5) | 1.00 | 1.00 |
| Parity of the mother | 21(13.5) | 71(19.5) | 2.00 (0.72–5.55) | 2.38(.75–7.53) |
| Prim parous (1) | | | | |
| Multipara (2–4) | 115(73.7) | 236(64.8) | 2.89 (1.27–6.59) | .87(.35–2.18) |
| Grand multipara (5+) | 20 (12.8) | 57(15.7) | 1.00 | 1.00 |
| No of children | | | | |

(*Continued*)

**Table 4.** (Continued)

| Variables | Complementary feeding | | COR (95% CI) | AOR (95% CI) |
|---|---|---|---|---|
| | Appropriate | Inappropriate | | |
| Maternal education | No (%) | No (%) | | |
| 1–3 | 125(80.1) | 259(71.2) | 1 | 1 |
| 4–6 | 25(16) | 95(26.1) | 81(.29–2.29) | 1.34(.34–5.13) |
| >6 | 6(3.8) | 10(2.7) | 1.28(.44–3.73) | 1.63(.50–5.27) |
| Birth interval | | | | |
| ≤2 | 103(66) | 235(64.6) | 1 | 1 |
| 3–4 | 45(28.8) | 115(31.6) | 1.57(1.07–2.32) | .97(.36–2.63) |
| >4 | 8(5.1) | 14(3.8) | 1.22(.50–2.99) | 1.52(.57–4.06) |
| House hold Wealth index | | | | |
| Lowest | 1(0.5) | 12(3.6) | .68(.08–5.83) | .63(.07–5.56) |
| Second | 22(12) | 39(11.6) | 4.63(2.0–10.71)** | 4.27(1.8–10.22)** |
| Middle | 80(43.5) | 123(26.6) | 5.33(2.61–0.90)** | 4.02(2.23–9.94)** |
| Fourth | 71(38.6) | 80(23.8) | 7.28(3.51–5.11)** | 7.02(3.27–15.1)** |
| Highest | 10(5.4) | 82(24.4) | 1 | 1 |
| House Hold Food Security | | | | |
| Secured | 111(71.2) | 296(81.3) | 1.59(1.01–2.52)** | 1.71(1.07–2.75)** |
| Not Secured | 45(28.8) | 68(18.7) | 1 | 1 |
| Sources of information | | | | |
| TV | 17(10.9) | 60(16.5) | 1 | 1 |
| Radio | 36(23.1) | 61(16.8) | 2.08(1.06–4.10)* | 1.67(.77–3.65) |
| HCWs | 27(17.3) | 27(7.4) | 3.53(1.65–7.53)** | 3.5(1.45–8.26)** |
| Reading | 55(35.3) | 182(50) | 1.07(.58–1.98) | 1.02(.49–2.08) |
| Relative/family | 21(13.5) | 34(9.3) | 2.18(1.01–4.68)* | 2.36(.96–5.77) |
| Mothers attitude | | | | |
| Positive | 108(69.2) | 283(77.7) | 1.55(1.02–2.36)* | 1.80(1.15–2.82* |
| Negative | 48(30.8) | 81(22.3) | 1 | 1 |
| Mothers' Knowledge | | | | |
| good Knowledge | 107(68.6) | 212(58.2) | 1.57(1.05–2.33)* | 2.34(1.43–3.84)** |
| poor knowledge | 49(31.4) | 152(41.8) | 1 | 1 |

* = stands for value of ≤0.05

** = stands for P. vale of less than or equal to 0.001, TV: television, HCWs: health care workers, ANC: Antenatal care, PNC: Postnatal care, ETB: Ethiopian birr, COR:, crud odd ratio, AOR: Adjusted Odd ratio, CI: confidence interval.

**Note**: Table 4: Factors associated with complementary feeding of children age 6–23 months.

The findings revealed factors independently and positively associated variables in adjusted analysis were socioeconomic status (wealth index, household food security, and household income), sources of information, child age. After controlling for potential confounders, the number of ANC follow-ups, mothers' knowledge, and positive attitude are factors significantly positively associated with ACFP, whereas the remaining variables were not associated after controlling for potential confounders, despite being associated in bivariate analyses.

Maternal ANC follow-up status was one of positive indicators for appropriate complementary feeding practice. Only antenatal care follow-up of four times or more was found to be statistically significantly positively associated with appropriate complementary feeding practice; mothers who had ANC $N^o \geq 4$ times were 2.6 times more likely to practice appropriate complementary feeding than those counterpart (AOR = 2.57, 95% CI: 1.3, 5.05). The

socioeconomic status of the household (wealth index and food security) was found to be statistically significantly positively associated with complementary feeding practices. Mothers in the second, middle, fourth, and highest percentiles were approximately four times more likely to practice appropriate complementary feeding than those in the lowest percentile (AOR = 4.27, 95 percent CI: (1.8–10.22) **, 4.02, 95 percent CI: (2.23–9.94) **, and 7.0, 95 percent CI: (3.27–15.10) **, respectively.

The other positively associated variable on appropriate complementary feeding practice was adequate food security. After controlling for potential confounders, our findings show that mothers who lived in a household with adequate food security were 1.7 times more likely to practice appropriate complementary feeding than mothers who lived in an insecure household (AOR = 1.71,95 percent, CI: (1.07–2.75 **)).

The other positive indicator with appropriate complementary feeding practice was advice on breast feeding from health workers. Children whose mothers received breast feeding information from health workers were 3.5 times more likely to practice ACFP than children whose mothers received information from other sources (AOR = 3.5,95 percent, CI: 1.45–8.26) **).

Mothers with children aged 12–17 months and 18–23 months showed a positive association in practicing ACFP compared to other groups of children [AOR = 2.32,95 percent, CI:(1.40–3.82) **.and AOR, = 95, CI:1.91(1.10–3.32)] **. Result also revealed that knowledge and attitude about ACFP of the mothers was significantly associated with ACFP. Children from households where the mother having a positive attitude were 1.8 times more likely to practice ACFP than those from households where the mother had a negative attitude (AOR = 1.80, 95 percent CI = (1.15, 2.82)) *. Similarly, the likelihood of practicing ACFP among mothers having good knowledge was 2.34 times higher than those mothers who had poor knowledge [AOR = 2.34,95 percent, CI: (1.43–3.84)** (See Table 4).

## Discussion

The current study, which had a response rate of 97.01 percent, was conducted on the magnitude and factors associated with appropriate complementary feeding practice (ACFP). Overall, the magnitude of ACFP was 30%, which is higher than the study done in Negele Arsi, 9.5 percent, Damota Sore 11.4 percent, and Ghana 14.3 percent [13–15], but lower than the Sri Lankan 68 percent, Bangladeshi 40%, and Nepal 32 percent [9, 10]. This could be due to differences in study setting, socioeconomic status, or indicators used to measure appropriate complementary feeding, as well as sociocultural differences between different populations at different times.

In a recent study, the lowest proportion of children who had been fed ACFP was in the age group of 6-11months when compared to the counter group, implying the importance of paying special attention to the younger age group. In this study, the proportion of children aged 6–23 months who had been introduced to solid, semi-solid, or soft foods was 67.9 percent, while the proportion of children aged 6–8 months was 86/93. (92.5 percent).

This is higher than the 72.5 percent reported by Nagele Arsi, 74.2 percent by Damota Sore, 72.6 percent by Ghana, and 70–71 percent reported by Nepal and Bangladesh studies [9, 10–15] This figure corresponds to the WHO recommendation that more than 80% of 6–8-month-old children begin complementary feeding at 6 months of age [5, 6]. Nonetheless, the time of introduction of complementary feeding in the current study is better than in other similar studies [9, 10, 13–16].

Maternal health care services such as ANC, PNC, and institutional delivery services utilization, as well as extensive effort of HEWs, health development army, and information differences in the study area, were better in the current study, which could result in better awareness

and practices on appropriate time of complementary feeding introduction when compared to other studies. Other explanations include differences in time dimension and socio-cultural diversity in the study setting.

The proportion of 6–23-month-old children who met the MMF criteria was (61.7 percent). It is nearly comparable to the findings of Nagle Arsi (67.3%) and Bale Zone Ethiopia (68.4%) [13, 19]. However, it is lower than studies conducted in Sri Lanka (88.3 percent), Bangladesh (81%), Nepal (82%), coastal South India (77.5 percent), Derashe, Southern Ethiopia (95%), and Amibara district, North East Ethiopia (69.2%) [9, 10, 20–23]. This disparity could be attributed to caregivers' sociocultural, educational, and working conditions.

Regarding minimum dietary diversity (MDD), the current study also revealed that 42.5 percent of children met MDD, which reflects as only this mothers fed their young child with four or more food groups from seven food sets, i.e. grains, roots and tubers; legumes and nuts; dairy products; flesh foods; vitamin A reached food; eggs, and other fruits and vegetables, which is almost similar to that of Bangladesh 42 percent, but higher than the fig stated from studies done in, Nagle Arsi 18.8%, Damota sore 16%., India 15%, Nepal 34%, however it's lower than that of Sir Lanka 71% [9, 10, 13, 14]. In Eastern and Southern Africa, one out of every ten infants and young children died [8].

The high variation from Damota sore and Nagle Arsi could be attributed to the fact that the current study was conducted in the city, where there was better access to information and maternal health care, resulting in differences in maternal awareness and educational status variations, whereas previous studies were conducted in rural areas, where mothers were less advantageous compared to their urban counterparts. Low consumption of protein-rich foods can be attributed to a variety of factors, including a lack of nutritional awareness and a lack of access due to economic constraints [24, 25].

The percent study revealed that the variety of foods given to younger children is lower, and MDD only tends to increase with increasing age (Table 3), i.e. the lowest proportion of children who met MDD was found in the age group of 6–11 months (Fig 4). Identical patterns have been observed in Ethiopia and other developing countries [13, 20, 26]. This could be because mothers believe that younger infants do not require a diverse diet, or because their guts are unable to digest animal-sourced foods. Aside from that, the most commonly restricted foods are meat and cabbage, leading mothers to believe that their children cannot be swallowed.

Furthermore, flesh food is the least consumed food across all age groups, with eggs being the least consumed in the age group 6–11 months. As a result, CF may be started with monotonous staples. This is consistent with the findings of a Northern Ethiopian study, which discovered that flesh foods and eggs were introduced into children's diets in the middle of the second year of life [20]. The study discovered that household economic status, as measured by the wealth index and level of food security, was a significant predictor of MDD. Obviously, the household's lower economic status limits the availability and variety of food. The minimum acceptable diet, which included MMF and MDD, was 41.7 percent. This is comparable to findings from Bangladesh (40%) and Ghana (46%), both of which have MAD [9, 10, 15]. However, this is higher than the national level of Ethiopia (7%) and Abiy Adi, North Ethiopia (11.9%), India (9%), Nepal (32%), however, it was lower than Sirlanka's finding of 68 percent [9–11, 13–15, 20, 27].

The lower level of the result could be attributed to socioeconomic, cultural, and policy differences between the study area and time. As a result of the low prevalence of this indicator, the majority of children were either not fed as frequently as the recommended 2–4 times daily, or were not offered food from four or more of the recommended food groups in their diet,

which may have resulted in inappropriate CF practices that resulted in malnutrition. In this study, high MAD was observed in comparison to the national figure of 7% as of EDHS 2016.

This could be because EDHs were a nationally representative survey with a wide range of child feeding styles in different parts of Ethiopia with diverse sociocultural contexts, and the DHS covers both rural and urban areas, which reduces the figure. The higher figure observed in our study could be attributed to the current expansion of HEWs in the study area, which focuses on ANC, PNC, and child care education, increasing maternal exposure to healthcare workers and thus increasing their practices. Furthermore, this is a pocket study that is localized in a town where there is better access and availability to information, education, health care service utilization, and other social services over time.

The proportion of children who met the ACFP was 30% overall. This was higher than the results from Negele Arsi (9.5%), Damota Sore (11.4%), and Abiy Adi town in North Ethiopia, where ACFP was 10.5 percent [13, 14, 20]. This could be due to differences in study settings, as the previous study was conducted in rural areas of the country where access to maternal health care services and media is limited.

## Factors associated with appropriate complementary feeding practices

The current study found that household socioeconomic status (wealth index and food security status, household income), sources of information, number of ANC follow ups, child age good knowledge, and positive attitude are factors that are significantly positively associated with ACFP, whereas the rest of the variables were not associated or lost association after controlling for potential confounders, despite being associated in bivariate analyses.

Only ANC follow-up of four times or more was found to be statistically significantly positively associated with ACFP, i.e. mothers who had four or more ANCs were 2.6 times more likely to practice ACFP than their counterparts (AOR = 2.57, 95 percent CI: 1.3, 5.05[**]). This finding is consistent with findings from Damota Sore, Sri Lanka, Nepal, and India, where fewer antenatal clinic visits were associated with a delay in the introduction of complementary food [9, 10, 14, 15]. ANC contacts$\geq$ 4 times were a significant predictor of ACFP. This could be due to the effects of information and counseling provided by health care providers to mothers during their antenatal care services utilization. This is demonstrated by the fact that mothers who have contact with health care workers are significantly associated with ACFP (AOR = 3.5, 95% CI: (1.45–8.26) [**]). This could imply that promoting the use of maternity services and a stronger integration with IYCF aids in the improvement of infant feeding practices.

Household economic status, as measured by the wealth index and food security level, was another important determinant factor positively associated with ACFP. The Household Wealth Index was discovered to be statistically significant associated with CF practices.

Mothers in the second, middle, and fourth percentiles were about four times more likely to practice ACFP than those in the lowest percentile (AOR = 4.27, 95 percent CI: (1.8–10.22) [**], 4.02, 95 percent CI: (2.239.94) [**], and 7.0, 2.95 percent CI: (3.27–15.10) [**], respectively. As a result, this demonstrates that people with higher incomes are more likely to practice ACFP than their counterparts. Indeed, a higher household wealth index is associated with a higher dietary diversity score (DDS).

Families in the top percentile are more likely to be able to afford and provide a variety of foods to their children on a regular basis. Previous research in Sri Lanka, Pakistan, India, Bangladesh, and Nepal [9, 10] has consistently found a positive relationship between households with the highest wealth percentile and increased diet diversity. This was consistent with what we discovered.

The global nutrition report 2018 also revealed that infant and young child diets are suboptimal across all wealth groups, ranging from 75.6 percent in the lowest quintile to 56.7 percent in the highest quintile [27]. The fact that household wealth predicts MDD emphasizes the importance of household assets in determining optimal CF practices [9, 10], which is consistent with our findings.

This study contradicted the findings of Nagele Arsi, Damota Sore Ethiopia, Nepal, and Sirlanka, in which maternal education was found to be a predictor of ACFP [9, 10, 13, 14]. Possible explanations include differences in societal norms and cultures, as well as geographic differences in female education.

Child age is also found to be a predictor variable, with older children (12–23 months) approximately three times more likely to be fed appropriately than younger children (6–11 months). Similarly, studies conducted in five Asian countries as well as Tanzania, Nagele Arsi, and the northern part of Ethiopia found child age to be a predictor variable [8–10, 13, 20, 24]. This may provide an opportunity for health planners to devote adequate attention to the feeding of younger children.

According to the Global Nutrition Report, 74.6 percent of children aged 6–23 months do not have sufficient diet diversity for a healthy diet worldwide, so inappropriate CF after 6 months of age is one major cause of malnutrition, and malnutrition is the leading cause of the global burden of disease, so attention should be paid to young child nutrition education intervention to meet SDGs. Agnda refers to the abolition of "all forms of malnutrition" [28].

Attitude was another important predictor variable. Positive maternal attitudes toward ACFP are statistically significant. This could be due to the counseling the mothers received from health care providers during their use of antenatal care services. The most important issues regarding feeding during illness were only 63 (12%) mothers increasing food quantity and frequency. Despite cultural and social food restrictions, cabbage and meat are two foods that require special attention.

Another important variable was mothers' knowledge, with those with good knowledge of ACFP outperforming those with poor knowledge, according to a study conducted in Tehran [29]. This could be due to the information provided during the ANC attendant, which accounts for 95% of the data in this study.

Interestingly, encouraging practices were discovered in this study:—almost all mothers 519 (99.8 percent) breast feed their children after delivery, 496/520(95.5 percent) initiated Breast Feed earlier, and the majority of them 482(92.7 percent) feed breast milk based on child demand, whereas 489(94 percent) fed EBF versus 58 percent national figure of 2016 EDHS. It is recommended that a child breastfeed until the age of two years.

However, according to the EDHS 2016, the percentage of children who are currently breastfeeding drops from 91 percent among children aged 12–17 months to 76 percent among children aged 18–23 months in Ethiopia, which is somewhat comparable to this finding.

Homemade food was the most commonly preferred food for the child among 398 (76.5 percent), while the remaining 122 (23.5 percent) did not. On the other hand, thirty-one (6%) mothers started CF earlier, and breast-fed continuation was not as recommended at 1&2 years / I.e. not satisfactory, but far better than Nigeria and Kosovo [30, 31]. Mothers' negative attitude toward the quantity of breast milk, failure to stay with the child, and a lack of knowledge about breast milk were the most common reasons for early initiation of CF.

Almost 488 (86.2%) mothers used bottle feeding, whereas during illness, only 63 (12%) mothers increased food quantity and frequency, while the remaining 88 percent did not. Despite cultural and social food restrictions, the most commonly consumed foods are cabbage and meat, which account for 124 percent of total consumption (23. 8 percent). 452 (86.9%) prepare food separately, while 68 (13.1%) prepare with adult food. Overall, low levels of ACFP,

MDD, and MAD were observed in this study, particularly among children aged 11 months. As a result, mitigation is required to improve CF.

## Limitations

Because it is cross-sectional and collects point-in-time data, it does not show a cause-and-effect relationship. Because the study was conducted in the town, it is possible that the results will be exaggerated. Furthermore, the figure may have been overestimated or underestimated due to recall and social desirability biases that may have been introduced to the time of initiation, diversity, and frequency of food. The 24-hour dietary diversity recall may only show the most recent feeding and requires repeated measurements.

Another limitation of our study is that it was conducted on a single town's population, which may not be representative of the region. Even when all confounding factors are known and controlled for using conventional multivariate analysis, the observed association between exposure and outcome can still be dominated by residual confounding effects. Therefore, an observed significant association apparently provides limited evidence for a causal relationship.

## Conclusion

As a result, the overall prevalence of ACFP was low, negatively impacting the health of infants and young children. This demonstrates the importance of taking immediate action to promote ACFP.

Despite the fact that complementary feeding was introduced at a high rate, the proportion of mothers who met MMF, MDD, and MAD was also low. ACFP was significantly associated with child age, 4 ANC follow up, household socioeconomic status, sources of information, mothers' knowledge, and positive attitude.

## Recommendation

Previous studies reported that inappropriate complementary feeding was responsible for two-thirds of child deaths; however, ACFP was low in this study. So, to scale up these successful interventions to meaningful levels the Health Bureau, non-governmental organizations, and other development sectors should pay special attention to:

- Nutritional education/counselling for mothers and/or caregivers is critical for improving infant and young child feeding practices, particularly for mothers with younger children, regarding time, variety, quantity, and frequency of food.

- Promoting the socioeconomic status of the community, particularly poor mothers with low wealth index households/food insecurity, through multidisciplinary/inter-sectoral collaboration engagement to improve ACFP.

- HCWs should encourage women to attend more ANC and pay special attention to mothers during infant nutrition counseling/educational interventions in order to improve complementary feeding practices and reduce malnutrition.

- A large-scale longitudinal study will be proposed to the researcher in order to arrive at the true figure.

## Supporting information

**S1 File.**
(ZIP)

## Acknowledgments

The investigators would like to thanks Madda Walabu University, Shashemene Campus for offering support they had given us, Materials used in this project are honorable acknowledged. The investigators express their sincere appreciations to peoples and organization (Shashemene Health Bureau, Shashemene City Administration for offering necessary information that help to conduct the research.

Lastly, we would like to present our heartfelt thanks to data collectors and study participant for support they provided us throughout the whole process of data collection period.

## Author Contributions

**Conceptualization:** Junayde Abdurahmen Ahmed, Kebede Kumsa Sadeta.

**Formal analysis:** Kebede Kumsa Sadeta, Kelil Hussein Lenbo.

**Investigation:** Junayde Abdurahmen Ahmed, Kelil Hussein Lenbo.

**Methodology:** Junayde Abdurahmen Ahmed, Kebede Kumsa Sadeta, Kelil Hussein Lenbo.

**Project administration:** Junayde Abdurahmen Ahmed.

**Software:** Kebede Kumsa Sadeta.

**Supervision:** Junayde Abdurahmen Ahmed, Kebede Kumsa Sadeta.

**Writing – original draft:** Junayde Abdurahmen Ahmed.

**Writing – review & editing:** Kebede Kumsa Sadeta, Kelil Hussein Lenbo.

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
