## [Decision Letter · Decision Letter 0]

20 Oct 2021

PONE-D-21-10678

Magnitude And Factors Associated With Appropriate Complementary Feeding Practice Among Mothers Having Children 6–23 Months Of Age In Shashemene Town, Oromia- Ethiopia; A Community Based Cross Sectional Study

PLOS ONE

Dear Dr. kumsa,

Thank you for submitting your manuscript to PLOS ONE. After careful consideration, we feel that it has merit but does not fully meet PLOS ONE’s publication criteria as it currently stands. Therefore, we invite you to submit a revised version of the manuscript that addresses the points raised during the review process.

We look forward to receiving your revised manuscript.

Kind regards,

Waliou Amoussa Hounkpatin, Ph.D.

Academic Editor

PLOS ONE

Journal Requirements:

3. Please include additional information regarding the survey or questionnaire used in the study and ensure that you have provided sufficient details that others could replicate the analyses. For instance, if you developed a questionnaire as part of this study and it is not under a copyright more restrictive than CC-BY, please include a copy, in both the original language and English, as Supporting Information.  If the original language is written in non-Latin characters, for example Amharic, Chinese, or Korean, please use a file format that ensures these characters are visible.

4. During your revisions, please confirm whether the wording in the title is correct and update it in the manuscript file and online submission information if needed. Specifically, please correct the title on the pdf.

5. Please update your submission to use the PLOS LaTeX template. The template and more information on our requirements for LaTeX submissions can be found at http://journals.plos.org/plosone/s/latex.

6. Thank you for stating the following financial disclosure: 

7. Thank you for stating the following in the Acknowledgments Section of your manuscript: 

"The investigators would like to thanks Madda Walabu University, Shashemene Campus for

offering the fund to carry out this research project that is targeted to find out community

problems. Materials used in this project are honorable acknowledged. The investigators express

their sincere appreciations to peoples and organization (Shashemene Health Bureau,

Shashemene City Administration for offering necessary information that help to conduct the

research.

Lastly, we would like to present our heartfelt thanks to data collectors and study participant for

support they provided us throughout the whole process of data collection period. "

"The investigators would like to thanks Madda Walabu University, Shashemene Campus for

offering the fund to carry out this research project that is targeted to find out community

problems."

8. In your Data Availability statement, you have not specified where the minimal data set underlying the results described in your manuscript can be found. PLOS defines a study's minimal data set as the underlying data used to reach the conclusions drawn in the manuscript and any additional data required to replicate the reported study findings in their entirety. All PLOS journals require that the minimal data set be made fully available. For more information about our data policy, please see http://journals.plos.org/plosone/s/data-availability.

9. PLOS requires an ORCID iD for the corresponding author in Editorial Manager on papers submitted after December 6th, 2016. Please ensure that you have an ORCID iD and that it is validated in Editorial Manager. To do this, go to ‘Update my Information’ (in the upper left-hand corner of the main menu), and click on the Fetch/Validate link next to the ORCID field. This will take you to the ORCID site and allow you to create a new iD or authenticate a pre-existing iD in Editorial Manager. Please see the following video for instructions on linking an ORCID iD to your Editorial Manager account: https://www.youtube.com/watch?v=_xcclfuvtxQ

10. Please ensure that you refer to Figure 2 in your text as, if accepted, production will need this reference to link the reader to the figure.

11. "Please upload a copy of Figure 4, to which you refer in your text on page 15. If the figure is no longer to be included as part of the submission please remove all reference to it within the text.

12. Thank you for submitting the above manuscript to PLOS ONE. During our internal evaluation of the manuscript, we found significant text overlap between your submission and the following previously published works, some of which you are an author.

- https://bmcnutr.biomedcentral.com/articles/10.1186/s40795-017-0202-y

- https://bmcpediatr.biomedcentral.com/articles/10.1186/s12887-016-0675-x

- https://www.hindawi.com/journals/jnme/2019/2869424/

Please revise the manuscript to rephrase the duplicated text, cite your sources, and provide details as to how the current manuscript advances on previous work. Please note that further consideration is dependent on the submission of a manuscript that addresses these concerns about the overlap in text with published work.

Reviewers' comments:

Reviewer's Responses to Questions

**Comments to the Author**

1. Is the manuscript technically sound, and do the data support the conclusions?

Reviewer #1: Partly

Reviewer #2: No

2. Has the statistical analysis been performed appropriately and rigorously? 

Reviewer #1: Yes

Reviewer #2: No

3. Have the authors made all data underlying the findings in their manuscript fully available?

Reviewer #1: Yes

Reviewer #2: No

4. Is the manuscript presented in an intelligible fashion and written in standard English?

Reviewer #1: Yes

Reviewer #2: No

5. Review Comments to the Author

Reviewer #1: Well-done on your submission. Please find below my comments:

1) Please avoid using abbreviations without explanation in the abstract. Always define abbreviations at first mention.

2) Please provide global or country wide estimate of complementary feeding practice.

3) Why Ethiopia?

"Ethiopia is one of the sub Saharan African countries with high level of malnutrition and has launched the national strategy for infant and young child feeding in 2004 to improve the nutritional status of children". This is all that is written about Ethiopia in the introduction which is not enough reason to conduct a study on complementary feeding in Ethiopia.

4) Please provide a proper literature review on complementary feeding practice globally and in Ethiopia. What does existing literature say about the prevalence and factors associated with complementary feeding practice? Why is it a topic of concern?

5) The buildup to the study rationale is weak.

6) A proper definition of the independent variables would be appreciated. For instance, what do you mean by food insecurity? How is it derived?

7) How was the effect of residual confounding as a result of unmeasured co-variates handled?

8) How was the dependent variable expressed in the analysis?

9) How was statistical bias account for and avoided?

10) What are the policy implications of your study?

11) Please check your punctuation and grammar.

Reviewer #2: The purpose of the study is interesting but not innovative. it is a theme that has been widely addressed in the literature.

The level of English is low and requires extensive proofreading and editing.

Additional statistical analysis needs to be done and results presented in a succinct manner using adequate format table.

The authors did not conduct a sufficient literature review on the topic and this makes the discussion superficial.

References are not well presented. The formats of the tables are mostly inappropriate. The font of the body text is not uniform. In total, this is an article carelessly written both on the content and on the form.

6. PLOS authors have the option to publish the peer review history of their article (what does this mean?). If published, this will include your full peer review and any attached files.

Reviewer #1: No

Reviewer #2: No

---

## [Author Response · Author response to Decision Letter 0]

9 Jan 2022

we have revised the document based on the request of the reviewer

---

## [Editor Report · Decision Letter 1]

18 Feb 2022

PONE-D-21-10678R1Magnitude And Factors Associated With Appropriate Complementary Feeding Practice Among Mothers Having Children 6–23 Months Of Age In Shashemene Town, Oromia- Ethiopia; A Community Based Cross Sectional StudyPLOS ONE

Dear Dr. kumsa,

Thank you for submitting your manuscript to PLOS ONE. After careful consideration, we feel that it has merit but does not fully meet PLOS ONE’s publication criteria as it currently stands. Therefore, we invite you to submit a revised version of the manuscript that addresses the points raised during the review process.

We look forward to receiving your revised manuscript.

Kind regards,

Waliou Amoussa Hounkpatin, Ph.D.

Academic Editor

PLOS ONE
---

## [Author Response · Author response to Decision Letter 1]

3 Mar 2022

we have sent the response to the reviewer request

---

## [Editor Report · Decision Letter 2]

8 Mar 2022

Magnitude And Factors Associated With Appropriate Complementary Feeding Practice Among Mothers of Children 6–23 Months Age in Shashemene Town, Oromia- Ethiopia: Community Based Cross Sectional Study.

PONE-D-21-10678R2

Dear Dr. kumsa,

We’re pleased to inform you that your manuscript has been judged scientifically suitable for publication and will be formally accepted for publication once it meets all outstanding technical requirements.

Kind regards,

Waliou Amoussa Hounkpatin, Ph.D.

Guest Editor

PLOS ONE

Additional Editor Comments (optional):

We realized that you have take full consideration to highlly improve  the quality of your manuscript. However, it will be interesting that you take the two following comments into acccount in the last version of your manuscript.

- Line 171, Line 626, [Fig.1] : Write «complementary » NOT « complimentary »

- Harmonisation of 6-23 months. Example : Change « 6-24 months » with « 6-23 months » in line 61
---

## [Editor Report · Acceptance letter]

14 Mar 2022

PONE-D-21-10678R2 

Magnitude And Factors Associated With Appropriate Complementary Feeding Practice Among Mothers of Children 6–23 Months Age in Shashemene Town, Oromia- Ethiopia: Community Based Cross Sectional Study.   

Dear Dr. Sadeta:

I'm pleased to inform you that your manuscript has been deemed suitable for publication in PLOS ONE. Congratulations! Your manuscript is now with our production department. 

Kind regards, 

on behalf of

Dr. Waliou Amoussa Hounkpatin 

Guest Editor

PLOS ONE